# Role of Glucagon-like Peptide-1 Receptor Agonists (GLP-1RAs) in Patients with Chronic Heart Failure

**DOI:** 10.3390/biom15091342

**Published:** 2025-09-19

**Authors:** Pasqual Llongueras-Espí, Elena García-Romero, Josep Comín-Colet, José González-Costello

**Affiliations:** 1Advanced Heart Failure and Heart Transplant Program, Cardiology Department, Bellvitge University Hospital, L’Hospitalet de Llobregat, 08907 Barcelona, Spain; pllonguerasespi@gmail.com (P.L.-E.); jgonzalez@bellvitgehospital.cat (J.G.-C.); 2Bio-Heart Cardiovascular Diseases Research Group, Bellvitge Biomedical Research Institute, L’Hospitalet de Llobregat, 08908 Barcelona, Spain; jcomin@bellvitgehospital.cat; 3Biomedical Research Networking Center on Cardiovascular Diseases (CIBER-CV), Carlos III Health Institute, 28029 Madrid, Spain; 4Community Heart Failure Program, Cardiology and Internal Medicine Department, Bellvitge University Hospital, L’Hospitalet de Llobregat, 08907 Barcelona, Spain; 5Department of Clinical Sciences, School of Medicine, University of Barcelona, 08036 Barcelona, Spain

**Keywords:** GLP-1 receptor agonists, heart failure, cardiometabolic disease, type 2 diabetes, obesity

## Abstract

Glucagon-like peptide-1 receptor agonists (GLP-1RAs) are widely used in the management of type 2 diabetes and obesity due to their metabolic benefits. Beyond weight loss and glycemic control, emerging evidence suggests they may also exert cardioprotective effects. In the context of heart failure (HF), particularly HF with preserved ejection fraction (HFpEF), GLP-1RAs have been associated with improvement in symptoms, physical capacity, biomarkers, and structural cardiac remodeling. These benefits appear to be independent of weight loss, suggesting additional mechanisms including anti-inflammatory effects, improved myocardial metabolism or modulation of epicardial adipose tissue. However, current data largely come from non-HF dedicated trials, with limited standardization of the HF phenotype. Results are overall inconsistent and may suggest potential harm in some cases, particularly in HF with reduced ejection fraction (HFrEF). This review aims to summarize the current evidence on the role of GLP-1RAs in heart failure, explore possible underlying mechanisms and highlight key gaps in knowledge.

## 1. Introduction

Heart failure (HF) is a progressive syndrome with diverse clinical manifestations, often resulting in considerable morbidity and mortality. Epidemiological data show a stabilization or decline in the incidence in high-income countries. However, its prevalence continues to increase due to better pharmacological treatments and therefore improved survival [1,2].

Current clinical guidelines divide HF into two different phenotypes: HF with reduced ejection fraction (HFrEF) and HF with preserved ejection fraction (HFpEF), since these two entities have different prevalence rates, incidences, treatments and prognoses.

HFrEF is characterized by impaired systolic function, typically defined as a left ventricular ejection fraction (LVEF) < 40%. It is often associated with ischemic cardiomyopathy or dilated cardiomyopathy, leading to progressive ventricular remodeling, reduced cardiac output, and symptomatic congestion. HFrEF has been extensively studied, and multiple therapies have demonstrated clear benefits in reducing hospitalization and mortality in this population.

In contrast, HFpEF is defined by an LVEF ≥ 50%, with symptoms of heart failure occurring despite preserved systolic function. HFpEF is frequently associated with aging, hypertension, obesity, and metabolic comorbidities, including type 2 diabetes. Unlike HFrEF, HFpEF lacks therapies with proven mortality benefit, and management focuses primarily on symptom control and comorbidity optimization. The growing prevalence of HFpEF, combined with its heterogeneous pathophysiology, has made it a major focus of recent research and clinical trials [1].

Diabetes has proven to be a cornerstone in the development of HFpEF and HFrEF, with complex and multiple pathways resulting in adverse myocardial remodeling and muscle dysfunction. There is a two- to four-fold increased risk of HF among people with diabetes compared to those without [3]. Conversely, heart failure itself increases the risk of developing type 2 diabetes, likely due to shared mechanisms such as insulin resistance and systemic inflammation [4,5].

Likewise, obesity is associated with an increased risk of both the incidence and mortality of heart failure, with risk increasing progressively with adiposity, independent of diabetes status, as seen in large-scale observational studies. Each 1 kg/m^2^ increase in Body Mass Index (BMI) confers a higher risk for heart failure, even exceeding that conferred by the presence of diabetes alone [6,7].

Current guideline-directed medical therapy (GDMT) for heart failure differs substantially between HFrEF and HFpEF. In HFrEF, four drug classes have demonstrated prognostic benefit and are considered foundational: inhibitors of the renin–angiotensin–aldosterone system, β-blockers, mineralocorticoid receptor antagonists and sodium–glucose co-transporter 2 (SGLT2) inhibitors. In contrast, for HFpEF, no therapy has consistently shown mortality benefit, and management should largely focus on controlling symptoms and associated comorbidities such as hypertension, atrial fibrillation and especially diabetes and obesity [8].

More recently, Cardiovascular Outcome Trials of new diabetes treatments, such as SGLT2 inhibitors, have also showed positive results in terms of HF hospitalization and mortality, independent of HbA1c levels [9]. This finding highlights the clinical need to explore novel pharmacological strategies.

Glucagon-like peptide-1 receptor agonists (GLP-1 RAs) are incretin-based therapies that have consistently shown a reduction in major adverse cardiovascular events (MACE) across multiple randomized controlled trials (RCT). GLP-1RAs target underlying metabolic and inflammatory mechanisms which are not addressed by current heart failure treatments. However, their impact on HF outcomes remains more equivocal and less well-established [10].

This review aims to provide a comprehensive overview of the clinical and mechanistic evidence regarding GLP-1 receptor agonists in heart failure, and to identify key gaps in current knowledge.

### Literature Search Strategy

To frame this narrative review, we performed a structured literature search in PubMed/MEDLINE and the Cochrane Library up to May 2025. The following search terms were used in different combinations: “GLP-1 receptor agonists”, “heart failure”, “HFpEF”, “HFrEF”, “cardiovascular outcomes”, and “mechanisms”. We included randomized controlled trials, meta-analyses, relevant observational studies, and mechanistic reports. Additional references were identified by manual screening of bibliographies and current international guidelines. Only articles published in English were considered.

## 2. GLP-1 Physiology and Mechanism of Action of Its Analogues

GLP-1 is a small peptide produced in the gastrointestinal tract (mucosal L-cells), pancreatic islet α-cells and central nervous system (nucleus of the solitary tract) [8]. It is rapidly deactivated by local gut and systemic dipeptidyl peptidase-4 (DPP-4) expression and subsequently eliminated from circulation via kidneys; being this the explanation of its short half-life and low plasma levels [9].

GLP-1 receptors (GLP-1 Rs) were initially detected in Islet ß-cells and the central nervous system (CNS). Further studies also identified these receptors in other organs and tissues such as pancreatic exocrine cells, the autonomic and enteric nervous system, blood vessels and Brunner’s glands [10]. The heart is not an exception, with GLP-1R being present in the four cardiac chambers and sinoatrial node [11].

GLP-1RAs are analogues of endogenous GLP-1 that have been structurally modified to impede its hydrolysis thus expanding its half-life [8]. By activating GLP-1R they stimulate insulin secretion, inhibit glucagon release, delay gastric emptying, and reduce appetite through central nervous system signaling [12].

The pleiotropic presence of these receptors also contribute to several other mechanisms of actions initially less well-defined but also remarkably important. These effects involve mechanisms related to microcirculation, endothelial function, systemic inflammation and cardiovascular function, among others [13]. A detailed description of the main functional characteristics of these analogues is provided below as well as in Figure 1:

### 2.1. Role of GLP-1 in Glycaemia

GLP-1 stimulates glucose-dependent insulin secretion, enhances proliferation and mitigates pancreatic β-cell apoptosis, maintaining their quality and functionality. Not only is its effect insulin-mediated, but it also inhibits glucagon release from α-cells, which subsequently inhibits hepatic gluconeogenesis. Downregulation of glucagon secretion further controls blood glucose levels through a negative feedback mechanism.

Moreover, effects on the gastrointestinal system such as gastric emptying delay and appetite decrease further play a role in reducing blood glucose production, which contributes to overall blood glucose control [11].

Subtle differences can be seen between GLP-RAs: long-acting agents (liraglutide, albiglutide, dulaglutide and semaglutide) seem to have a stronger glucose-lowering potential when compared to short-acting ones (exenatide, lixisenatide). The overall effect is about a 1–2% drop in HbA1c levels [10].

### 2.2. Role of GLP-1 in Body Weight

GLP-1 RAs were initially designed as a glucose-lowering therapy, but their effects on body weight made them seen as the key to treat diabetes as well as its surrounding comorbidities.

Owing to the CNS GLP-1Rs, these drugs have been seen to reduce appetite and food intake. Additionally, they influence satiety and thus reduce food consumption. The reason for these effects is found in the POMC/CART and NPY/AgRP neurons of the hypothalamus, which are the main ones responsible for hunger and fullness sensations, respectively [8].

A local and direct effect of GLP-1 in the stomach also plays a crucial role, reducing gastric motility and prolonging food retention. Furthermore, GLP-1 can slow down gastric emptying by activating the vagus nerve [9].

Recent meta-analyses demonstrated that this effect translated into sustained weight loss (from 6 to 18% depending on the GLP-1RAs), BMI reductions, and lower waist circumferences. Greater benefits have been found in younger women, people without diabetes, those with higher basal BMI, and longer treatment periods [10,11].

### 2.3. Role of GLP-1 in Lipid Profile

Whereas effect of GLP-1RAs is unequivocal when it comes to glucose-lowering potential or weight loss; less is known when it comes to lipid profile effect, probably due to the complexity and heterogeneity of its mechanism of action.

It has been seen to modulate hepatic intrinsic lipid metabolism and lipolysis, with some studies pointing out that it can reduce liver fat synthesis. It also modulates liver cholesterol metabolism and promote intracellular cholesterol homeostasis [12].

In recent clinical trials, there has been minor reductions in LDL-Cholesterol (range −0.08 to −0.16 mmol/L), total cholesterol (range −0.16 to −0.27 mmol/L) and triglycerides compared to placebo [13].

It remains uncertain whether these effects can be maintained over prolonged treatment periods. Taking into account the appetite-suppressing and satiety-inducing properties of GLP-1RAs, associated behavioral changes and weight loss may also play a role in improving lipid profiles, but this has not yet been formally studied [10].

### 2.4. Role of GLP-1 in Inflammation

GLP-1RAs have also been shown to reduce systemic inflammatory marker levels. This is of special interest given the relationship between inflammation and cardiovascular disease, and whether this could explain the cardioprotective effects of these drugs [14].

Recent studies have shown direct and indirect anti-inflammatory effects of these drugs. On the one hand, there are consistent reductions in expression and release of proinflammatory cytokines (IL-6, TNF-α), inhibition of NF-κB, a key factor involved in the regulation of inflammatory processes, reduced chemokine adhesion, modulation of macrophage activation, and inhibition of immune cell infiltration [15].

On the other hand, patients with obesity and T2D usually display low-grade systemic inflammation (thought to be secondary to ectopic and epicardial fat accumulation, hyperglycemia, and dyslipidemia). Strict control of these classical cardiovascular risk factors may also play a role in mitigating inflammatory response, albeit this complicates interpretation of the anti-inflammatory actions of GLP-1RAs [10].

Clinical data from STEP Trial (Semaglutide Treatment Effect in People with Obesity), which studied weight loss effects of Semaglutide, showed a consistent 30–40% reduction in C-reactive protein levels [16,17]. These findings are supported by further research and meta-analyses, showing reductions in C-reactive protein (a median of −0.54 mg/L) and TNF-α levels [18]. These anti-inflammatory markers are not associated with HbA1c levels, suggesting that their effects may be mediated through an independent mechanism of action.

### 2.5. Role of GLP-1 in Blood Pressure (BP)

Clinical studies have shown that there is a direct relationship between GLP-1RAs and blood pressure. Recent meta-analyses demonstrated that GLP-1RAs, especially short-acting GLP-1RAs, have a mild lowering BP effect. On average, there is a 2.4–5 mmHg reduction in blood pressure, depending on the type of GLP-1RAs and the baseline BP. This effect is sustained during treatment [19].

To date, particular pathways involved in this BP reduction are not yet fully understood. Preclinical studies show that GLP-1RAs can have a direct effect to the carotid body by diminishing its sympathetic excitability, with this being a promising target for managing cardiovascular metabolism [20].

Not only is this a possible target; GLP-1RAs have also been shown to affect in salt homeostasis and in atrial natriuretic peptide (ANP) production [21,22]. Other crucial factors that may play a role include weight loss and better glycemic control, albeit the blood pressure-lowering effects appear to be independent of HbA1c levels [23].

This effect, although mild, should not be underestimated, since randomized clinical trials have demonstrated that 5 mmHg systolic blood pressure reductions correlate with 10% decrease of major cardiovascular events, even at normal or high-normal blood pressure values [24].

### 2.6. Role of GLP-1 in Heart Rate (HR)

Studies in healthy volunteers, obese subjects and T2D patients have demonstrated that infusion of GLP-1RAs acutely increases HR [25,26]. This increase is on average about 2–4 bpm, as previously described in meta-analyses of 60 randomized controlled clinical trials with GLP-1RAs. This effect quickly wanes when the drug is eliminated [13].

Firstly hypothesized to be mediated by vasodilatation and reflex tachycardia, but recent studies described direct chronotropic effects on the heart mediated by receptors in the sinus node cells (stimulating calcium signaling via PKA dependent phosphorylation of Ca^2+^ cycling proteins) [27].

Subtle differences between GLP-1RAs have been noted, with a pronounced and prolonged increase in 24 h mean HR with long-acting agents, while short-acting ones lead to a more transient increase that reverts few hours after injection [28].

Although the deleterious effects of increased heart rate in cardiovascular patients -especially those with HFrEF- are well known [29], initial Cardiovascular Outcome Trials of GLP-1RAs did not report an increased cardiovascular risk in individuals with T2D at high cardiovascular risk [30,31,32]. However, in subjects with both T2D and advanced HF, the available but scarce data raise some concern [33].

### 2.7. Role of GLP-1 in Endothelial Function and Atherosclerosis

Endothelial cells constitute the inner surface of all blood vessels. The concept of ‘endothelial function’ is both diverse and complex at the same time. The endothelium is responsible for a variety of effects, involving vascular tone, vascular permeability, microcirculation and acting as organ-specific barriers, among others. The way of measuring its function must be, therefore, multiparametric [34].

Pathways involved in GLP-1RAs and endothelial function are heterogeneous. GLP-1RAs have been described to improve insulin-sensitivity, which enhances endothelial function, to activate nitric oxide (NO) synthase which produces vasodilatation and reduce VCAM-1 and ICAM-1 expression, mitigating local inflammation [35,36,37].

Coronary circulation has also been within the scope of these drugs. Directed studies have demonstrated an improvement in the coronary flow velocity reserve after 12 weeks of treatment with exenatide in T2D patients, which inversely correlated with ICAM-1 and VCAM-1 levels, highlighting its anti-inflammatory effects [38,39]. Other research has demonstrated a protective effect with intravenous infusion of GLP-1RAs in patients with coronary artery disease when assessed during dobutamine stress test [40].

The aforementioned effects have been shown to contribute to the stabilization of the atherosclerotic process. These include not only endothelial preservation and reductions in proinflammatory cytokines, but also improvements in lipid profiles, blood pressure control, and glycemic regulation, as previously described. Although the underlying mechanisms are diverse, they appear to converge towards a common cardioprotective pathway. This may help explain the observed reduction in major adverse cardiovascular events (MACE) among patients with type 2 diabetes and established cardiovascular disease treated with GLP-1RAs, as will be discussed further.

## 3. General Cardiovascular Effects of GLP-1RAs

Randomized clinical trials involving patients with T2D have demonstrated that strict glycemic control reduces the risk of microvascular complications. However, its effect on MACE remains less clearly established [41,42]. Subsequent studies revealed that certain glucose-lowering agents, particularly thiazolidinedione (pioglitazone) and DPP4 inhibitors (especially saxagliptin), could paradoxically increase the risk of heart failure events despite effective glycemic control [43,44,45], whereas other antidiabetic drugs, such as metformin, insulin, and sulfonylureas are largely neutral regarding HF outcomes. Consequently, careful agent selection in patients with or at risk for heart failure is mandatory [46,47].

These unexpected findings prompted regulatory agencies to mandate robust cardiovascular outcome data from RCT for all newly developed antidiabetic therapies—leading to the implementation of the so-called cardiovascular outcome trials (CVOTs).

Ironically, some of these agents, most notably the SGLT2 inhibitors, not only demonstrated cardiovascular safety but also showed significant cardiovascular benefit [9]. Similarly, GLP-1RAs have yielded comparable results:

### GLP-1RAs Cardiovascular Outcome Trials Results:

To this date, eight different RCT regarding GLP-1RAs and cardiovascular outcomes in T2D patients have been published [26,30,32,48,49,50,51,52]. All trials sought to determine whether GLP-R1As had a detrimental effect on cardiovascular outcomes.

Published between 2015 and 2021, these eight trials were (in order of when their primary results were reported): ELIXA (lixisenatide) [30], LEADER (liraglutide) [48], SUSTAIN-6 (semaglutide) [32], EXSCEL (exenatide) [49], Harmony Outcomes (albiglutide) [50], REWIND (dulaglutide) [51], PIONEER 6 (oral semaglutide) [26] and AMPLITUDE-O (efpeglenatide) [52].

All eight randomized controlled trials (RCTs) shared key methodological and clinical characteristics: all enrolled participants were adults (age > 18 years) with T2D and established or high/very-high cardiovascular risk. Notably, ELIXA exclusively enrolled patients who had recently experienced an acute coronary syndrome (ACS).

Each trial employed a double-blind, parallel-group design. Among them, PIONEER 6 was the smallest in terms of sample size (n = 3183) and had the shortest median follow-up duration (1.3 years). In contrast, EXSCEL randomized the largest number of participants (n = 14,752), while REWIND had the longest follow-up period (median 5.4 years). MACE was the primary endpoint for all trials, apart from ELIXA which also included hospital admission for unstable angina to the primary endpoint.

The general characteristics of the study populations showed a mean age ranging from 60 years (ELIXA) to 66 years (PIONEER 6 and REWIND). The majority of patients were white, with proportions ranging from 70 to 87% in Harmony Outcomes and AMPLITUDE-O respectively. Most of them were male, with the lowest proportion of males found in REWIND trial (54%).

All participants in the ELIXA and Harmony Outcomes trials had established cardiovascular disease at the time of randomization. In contrast, the remaining trials enrolled individuals with either established CV disease or at high CV risk. Of note, the REWIND trial included the lowest proportion of participants with established cardiovascular disease (n = 3114; 31%). Other basic characteristics and main results can be found in Table 1.

Overall, the eight CVOTs demonstrated a reduction in three-point MACE, primarily driven by a lower incidence of fatal or non-fatal stroke, along with a trend toward reduced rates of non-fatal myocardial infarction, cardiovascular death, and all-cause mortality. GLP-1 receptor agonists appeared to have little or no effect on hospitalization for heart failure, with the exception of AMPLITUDE-O (Efpeglenatide) and Harmony Outcomes (Albiglutide), which showed a statistically significant reduction in HF events. These cardiovascular benefits were more pronounced in patients with type 2 diabetes and established cardiovascular disease.

However, the interpretation of these findings was limited by relevant differences across the trials: different molecules (especially exendin-4 base GLP-1RAs vs. human-based GLP-1RAs) which could alter the results [48], different baseline cardiovascular risk and event rates, different baseline treatment (e.g. SGLT2 inhibitors) as well as differences in trial design and duration. Furthermore, hospitalization for heart failure was not a primary endpoint in most studies, and heart failure phenotyping was generally lacking, as will be discussed further.

Subsequent meta-analyses of eight CVOT helped clarifying these observations. Giugliano et al. [49] demonstrated that GLP-1RAs in T2D individuals reduced the risk of MACE by 14% (HR = 0.86, 95% CI, 0.79–0.94, *p* = 0.006) compared with placebo, driven by a reduction in all three components. There was a 13% reduction in the risk of CV mortality (HR = 0.87, 95% CI, 0.78–0.96, *p* = 0.016), fatal or non-fatal MI (HR = 0.91, 95% CI, 0.81–1.01, *p* = 0.078) and non-fatal stroke (HR = 0.84, 95% CI, 0.76–0.94, *p* = 0.007). All cause-mortality (HR = 0.88, 95% CI, 0.80–0.96, *p* = 0.012) and hospitalization for heart failure (HR = 0.90, CI 95%, 0.83–0.98, *p* = 0.023) were also significantly reduced. Individuals with prior CVD seemed to benefit more from this effect.

Kristensen et al. [53] and Sattar et al. [54], confirmed these findings. Especially the latter, who demonstrated that these effects were consistent across multiple subgroups. No significant differences in the outcomes were observed based on the presence or absence of established cardiovascular disease, baseline HbA1c level, median duration of follow-up, type of GLP-1RAs (human-based vs. exendin-4 based), BMI (<30 vs. >30 kg/m^2^), Age (<60 vs. >60 years) and baseline estimated GFR (<60 vs. >60 mL/min/1.73 m^2^).

Notably, this analysis also demonstrated that with the exclusion of the ELIXA trial, HRs led to an overall improvement across all outcomes. Treatment with GLP-1 receptor agonists was associated with a 15% relative risk reduction in MACE (HR 0.85; 95% CI 0.80–0.90; *p* < 0.0001), a 15% reduction in CV death (HR 0.85; 95% CI 0.78–0.93; *p* = 0.0005), a 12% reduction in fatal or non-fatal MI (HR 0.88; 95% CI 0.81–0.96; *p* = 0.0048), and a 19% reduction in fatal or non-fatal stroke (HR 0.81; 95% CI 0.74–0.90; *p* < 0.0001). The NNT to prevent one MACE outcome over a weighted average mean follow-up of 3.0 years was 65 patients.

In conclusion, in patients with T2D who are at a high/very-high cardiovascular risk GLP-1RAs have been found to reduce the incidence of major adverse cardiovascular events, including cardiovascular death, nonfatal myocardial infarction, and nonfatal stroke, as well as all-cause mortality and hospitalization for heart failure.

## 4. GLP-1RAs and Chronic Heart Failure (HF)

Alterations in cardiac metabolism—such as impaired fatty acid oxidation and myocardial insulin resistance—are recognized contributors to the pathophysiology of heart failure [55]. However, current heart failure therapies do not specifically target these metabolic abnormalities.

Preclinical studies have demonstrated that recombinant GLP-1 improves myocardial insulin sensitivity and confers cardioprotective effects during ischemic conditions [40]. In a pilot clinical trial, administration of recombinant GLP-1 was linked to enhanced cardiac function and increased exercise tolerance in individuals with advanced HF and reduced left ventricular ejection fraction (LVEF) [56].

However, although these findings and that the results from CVOTs have suggested a potential reduction in hospitalization for heart failure with GLP-1 receptor agonists—estimated at approximately 10–11% in recent meta-analyses [54,56]–the findings from dedicated trials have been less encouraging.

### 4.1. GLP-1RAs and Heart Failure with Reduced Ejection Fraction (HFrEF)

Only a few randomized trials have evaluated GLP-1RAs in patients with HFrEF. While some studies designed to assess GLP-1RAs effects on cardiac function or myocardial metabolism are encouraging, their results for clinical endpoints have been less conclusive and sometimes suggestive of harm. This uncertainty raises concerns about the role of GLP-1RAs in HFrEF and underscores the need for further dedicated research.

The FIGHT trial (Functional Impact of GLP-1 for Heart Failure Treatment) was a randomized controlled trial that included 300 patients with advanced heart failure and a recent hospitalization for decompensated HF. Participants were randomized to receive liraglutide or placebo for 180 days. The primary endpoint was a global rank score incorporating time to death, time to rehospitalization for HF, and change in NT-proBNP levels. No significant difference was observed between groups (*p* = 0.31). Additionally, liraglutide was associated with an increase in heart rate and a non-significant tendency towards higher rates of rehospitalization for heart failure in patients with T2D (*p* = 0.07) [57].

Another randomized trial including 82 patients with heart failure in NYHA class II or III and LVEF <40% failed to demonstrate significant differences in LVEF (2.4% ± 1.1% vs. 4.4% ± 1.1%; *p* = 0.22), 6-min walk distance (18 ± 12 m vs. 9 ± 11 m; *p* = 0.58), or peak oxygen consumption after 12 weeks of treatment with albiglutide [58].

Finally, in the LIVE trial, 241 patients with reduced LVEF were randomized to receive either liraglutide or placebo for 24 weeks. No significant difference in LVEF was observed between groups (mean difference −0.8% [−2.1 to 0.5]; *p* = 0.24). However, patients treated with liraglutide experienced a higher incidence of serious cardiac events, including one death due to ventricular tachycardia (VT), non-fatal VT, atrial fibrillation requiring intervention, and one case of worsening heart failure [59].

These findings were subsequently supported by a post hoc analysis of the FIGHT trial, which demonstrated a consistent pattern of increased adverse events associated with liraglutide compared to placebo. This signal appeared to be primarily driven by a higher incidence of arrhythmias and worsening heart failure events, particularly among patients with more advanced HF (NYHA class III or IV symptoms) or those with concomitant type 2 diabetes [33].

Similarly, a prespecified subgroup analysis by baseline LVEF in the EXSCEL trial revealed opposing trends for hospitalization for heart failure (HHF). In participants with LVEF < 40%, once-weekly exenatide was associated with an increased risk of HHF (HR 1.52, 95% CI 0.95–2.43), whereas in those with LVEF ≥ 40% the effect appeared beneficial (HR 0.74, 95% CI 0.55–1.01), with a statistically significant interaction (*p*-interaction = 0.012). No LVEF-based effect modification was observed for MACE or mortality, and obesity did not alter the impact of once-weekly exenatide on HHF [60].

The latter findings align with the prior studies investigating GLP-1RAs in heart failure. Taken together, GLP-1RAs have been associated with a trend toward higher rates of HF hospitalization and all-cause mortality, as well as an increased incidence of adverse events in patients with advanced HFrEF, especially in NYHA class III-IV. This pattern was later confirmed for once-weekly exenatide in the LVEF < 40% and NYHA class III-IV. These findings suggest a potential detrimental effect in HFrEF and further underscore the limited evidence available in this phenotype.

Therefore, based on the current evidence, the use of GLP-1RAs in patients with established moderate to severe HF should be approached with caution. These adverse effects may be attributed to the ability of GLP-1RAs to increase heart rate and activate cyclic adenosine monophosphate (cAMP) signaling pathways, physiological mechanisms that could be harmful in individuals with HFrEF [61].

Several factors may explain the discrepancies observed between CVOTs and dedicated HF rials regarding the effects of GLP-1 receptor agonists (GLP-1RAs).

Firstly, none of the CVOTs included HF as a primary endpoint. Hospitalization for heart failure (HHF) was typically assessed as a secondary outcome, resulting in insufficient statistical power to detect meaningful differences in HF events.

Secondly, there was no standardized or universally applied definition of HF across these trials. Additionally, most CVOTs lacked comprehensive clinical characterization of HF, including data on symptom burden, NT-proBNP levels, echocardiographic findings, or concurrent HF-specific treatments.

Thirdly, patients with HF—particularly those with HFrEF—were underrepresented in these trials. Most of the HF population included appeared to have HFpEF, as illustrated in the EXSCEL trial, where LVEF data were available for 33% of participants and over 90% of them had an LVEF ≥ 40%.

These differences have prompted the design of dedicated RCT with the objective of evaluating the effects of GLP-1RAs in well-defined HF populations. In particular, recent studies have focused on specific heart failure phenotypes, especially HFpEF.

### 4.2. GLP-1RAs and Heart Failure with Preserved Ejection Fraction (HFpEF)

About 80% of patients with HFpEF also are overweight or obese [62]. Obesity has been widely recognized as a key contributor to HFpEF development and progression.

Visceral adiposity is a central known driver of both systemic and local inflammation, plasma and blood volume expansion, vasoconstriction, and increased epicardial and chest wall adiposity. These factors eventually lead to adverse cardiac remodeling with subsequent structural changes in ventricular structure and function (such as diastolic dysfunction, myocardial stiffness or left ventricular hypertrophy), which are hallmarks of HFpEF [7].

This has converted obesity in not only a comorbidity but a target to treat HFpEF and thus recent trials have focused on this population.

The first trial which explored this situation was the SELECT Trial, designed to assess semaglutide and cardiovascular outcomes in obese patients (BMI > 27 kg/m^2^) without diabetes and with established atherosclerotic cardiovascular disease [63]. While the primary endpoint focused on MACE, prespecified exploratory analyses examined a composite of HF–related outcomes which was defined as HF hospitalization, urgent medical visits or death from cardiovascular causes. In this study, there were 4286 HF patients, of whom 31.4% had HFrEF, 53% had HFpEF, and 15.5% had unclassified HF.

Semaglutide was associated with a significant 18% reduction in the overall risk of HF–related outcomes compared to placebo (HR 0.82; 95% CI, 0.68–0.98). However, when evaluating individual components, the reduction in hospitalization or urgent medical visits for heart failure did not reach statistical significance (HR 0.79; 95% CI, 0.60–1.03). An important limitation to bear in mind is that the trial did not clearly define HF phenotypes, who were adjudicated by the site investigator at the time of enrolment.

To address this limitations, two dedicated clinical trials were conducted: STEP-HFpEF and STEP-HFpEF DM (both of them including LVEF > 45% as an inclusion criteria).

In the STEP-HFpEF trial, 529 patients with obesity and HFpEF were randomized in a 1:1 ratio to receive either semaglutide 2.4 mg or placebo for 52 weeks. The co-primary outcomes were the change from baseline in the Kansas City Cardiomyopathy Questionnaire clinical summary score (KCCQ-CSS) and the percentage change in body weight. Secondary endpoints included a hierarchical composite outcome encompassing heart failure events [64].

The study showed significant improvements in both co-primary outcomes: the mean change in KCCQ-CSS was +16.6 points in the semaglutide group compared to +8.7 points in the placebo group, and the mean percentage reduction in body weight was −13.3% versus −2.6%, respectively, both being statistically significant.

While fewer HF events were observed in the semaglutide group vs. placebo (1 and 12 events respectively), the small number of events limited the statistical power, and thus this finding was considered exploratory.

Similarly, the STEP-HFpEF DM trial enrolled 616 patients with obesity, HFpEF, and T2D. Participants were randomized in a 1:1 ratio to receive either semaglutide 2.4 mg or placebo. Results mirrored those of the STEP-HFpEF trial, with a significantly greater improvement in KCCQ-CSS at 52 weeks in the semaglutide group (+13.7 vs. +6.4 points; *p* < 0.001) and a significantly larger percentage reduction in body weight (−9.8% vs. −3.4%; *p* < 0.001) [65].

In the STEP-HFpEF DM, adjudicated heart failure events (hospitalizations or urgent medical visits for HF) were less frequent in the semaglutide group compared to placebo (7 [2.3%] vs. 18 [5.9%] events; HR 0.40; 95% (CI 0.15–0.92), consistent with findings from the STEP-HFpEF trial. However, the absolute number of events remained relatively low, and this outcome should be interpreted with caution.

Taken together, the STEP-HFpEF and STEP-HFpEF DM trials demonstrated that once-weekly semaglutide 2.4 mg significantly improved HF-related symptoms, physical limitations, exercise capacity, and reduced body weight in patients with HFpEF—regardless of diabetes status.

Interestingly, a pre-specified pooled analysis of these two trials assessing the impact of baseline diuretic use showed that symptomatic improvement with semaglutide was more pronounced among patients already receiving loop diuretics. Moreover, loop diuretic dose was reduced by 17% in the semaglutide group, compared to a 2% increase in the placebo group (*p* < 0.0001). These findings suggest that the benefits of semaglutide extend beyond weight loss and may also involve modulation of other key metabolic mechanisms in HFpEF [66].

These results were further supported by a prespecified analysis of the FLOW trial, which demonstrated that people with T2D and chronic kidney disease treated with semaglutide had a significantly prolonged time to first HF events (HR: 0.73; 95% CI: 0.58–0.92; *p* = 0.0068) and CV death (HR: 0.71; 95% CI: 0.56–0.89; *p* = 0.0036); regardless of prior history of HF [67].

Similar to the SELECT trial, the FLOW trial was not specifically designed to evaluate heart failure outcomes, and therefore lacked systematic echocardiographic assessments and detailed HF phenotyping of all participants.

However, given the similarities across these four trials and the consistent trend toward heart failure (HF) benefits with semaglutide in HFpEF, Kosiborod et al. (2024) [68] conducted a pooled analysis which included 1914 patients with HFpEF randomized to semaglutide and 1829 to placebo. Semaglutide significantly reduced the risk of the composite endpoint of cardiovascular death or HF events (5.4% vs. 7.5%; HR 0.69, 95% CI 0.53–0.89; *p* = 0.0045), as well as the risk of worsening HF events (2.8% vs. 4.7%; HR 0.59, 95% CI 0.41–0.82; *p* = 0.0019). No significant effect was observed on cardiovascular death alone (3.1% vs. 3.7%; HR 0.82, 95% CI 0.57–1.16; *p* = 0.25) [68].

Importantly, the benefit appeared attenuated in patients with LVEF < 50% (representing 11% of participants—mainly derived from the SELECT and FLOW trials) or BMI < 35 kg/m^2^, suggesting that the observed effects may not extend uniformly across all HFpEF subgroups and should be interpreted with caution in these populations.

In a final noteworthy contribution, the SUMMIT trial was designed to further reinforce the emerging evidence on the role of GLP-1RAs in HF. The SUMMIT Trial randomized patients with HFpEF (defined as LVEF > 50%) and obesity (BMI > 30 kg/m^2^), assigning them to receive either tirzepatide or placebo. Tirzepatide significantly reduced the risk of the composite primary endpoint of cardiovascular death or worsening heart failure events (9.9% in the tirzepatide group vs. 15.3% in the placebo group; HR 0.62; 95% CI, 0.41–0.95; *p* = 0.026). This effect was primarily driven by a reduction in worsening heart failure events (8.0% vs. 14.2%; HR 0.54; 95% CI, 0.34–0.85), with no significant difference observed in cardiovascular mortality (2.2% vs. 1.4%; HR 1.48; 95% CI, 0.52 to 4.83) [69]. Table 2 summarizes the key features of these trials.

Of note, tirzepatide is not a pure GLP-1RA, but a dual GIP/GLP-1 receptor agonist, which implies a broader metabolic action that may enhance both glycemic control and weight reduction beyond what is typically seen with GLP-1RAs alone.

In conclusion, GLP-1RAs appear to improve HF-related symptoms, physical activity, exercise capacity and body weight. Interestingly, this overall beneficial effect seems to be independent of the latter, suggesting that additional mechanisms may be involved. GLP-1RAs have also been associated with a reduced risk of worsening heart failure events (including urgent medical visits and hospitalizations for heart failure).

This is supported by studies showing reductions in NT-proBNP levels among patients with HFpEF despite body weight loss (not seen in HFrEF). These findings are accompanied by favorable cardiac reverse remodeling, including reductions in LV mass and left atrial volume or improvements in diastolic function [69,70,71,72].

However, these beneficial effects cannot be confirmed in HFrEF, where the impact of GLP-1RAs remains unclear and may even be potentially harmful. Therefore, their use in this population should be approached with caution.

## 5. Potential Mechanisms of GLP-1RAs in Heart Failure

Based on the previous data and the findings from multiple clinical trials, the beneficial effects of GLP-1RAs in HF, particularly HFpEF, cannot be solely attributed to weight loss. Improvements in clinical, functional outcomes, even in the absence of significant weight reduction, alongside evidence of structural cardiac changes in echocardiographic studies, suggest that GLP-1RAs are cardioprotective drugs themselves in a weight-independent way.

Wang et al (2025), described multiple potential targets modulated by GLP-1RAs that might contribute to the cardiovascular outcomes previously described [73]. Among them is their effect on epicardial adipose tissue (EAT), which in excess is known to secrete proinflammatory adipokines, causing atrial and ventricular fibrosis secondary to deleterious local inflammation. This fibrotic remodeling contributes to increased myocardial stiffness and impaired diastolic function [74].

Beyond this local effect, chronic and low-grade systemic inflammation is also believed to play a central role in HFpEF, as observed in clinical trials showing elevated baseline elevated C-reactive protein levels in this population [64,75]. Notably, GLP-1RAs have demonstrated significant reductions in systemic inflammatory markers, though most studies are preclinical studies or trials in patients with T2D [18].

Other significant contributors to this so-called obesity-related HFpEF syndrome might be influenced by changes in myocardial energy metabolism, an effect already described in other drugs with cardiovascular benefits such as SGLT2 inhibitors [76,77]. GLP-1RAs enhance insulin sensitivity and thus improve cardiac metabolic efficiency. Although this metabolic shift is not as well-described as in SGLT2 inhibitors, recent evidence suggests a favorable shift toward a more efficient substrate use in the myocardium [9,78,79].

Finally, additional mechanisms that have been postulated to take part in these outcomes are renin–angiotension–aldosteron system (RAAS) inhibition and atherosclerosis delaying, both of which could contribute to the overall cardioprotective profile of GLP-1RAs [73].

Conversely, with respect to the potential adverse effects of GLP-1RAs in HFrEF, preclinical studies suggest that GLP-1RAs may influence cardiomyocyte survival through the AMP-activated protein kinase (AMPK) pathway. Activation of AMPK can modulate autophagy via downstream effectors, including mTOR signaling. While AMPK-driven autophagy is generally considered adaptive under acute stress, excessive or dysregulated autophagy may be maladaptive in the context of chronic HFrEF, potentially promoting cardiomyocyte loss and adverse ventricular remodeling. These mechanistic insights provide a plausible explanation for the trend toward increased HF–related adverse events observed in patients with HFrEF treated with GLP-1RAs [80].

Beyond AMPK-mediated autophagy, other mechanisms which may underlie the potentially adverse effects of GLP-1RAs in HFrEF include a sustained increase in heart rate (which could elevate myocardial oxygen demand) and enhanced sympathetic activity, all of which may exacerbate hemodynamic stress in a failing heart.

## 6. Gaps in Knowledge and Future Perspectives

GLP-1 receptors agonists have reshaped our understanding of cardiometabolic disease. Their benefits extend beyond traditional targets such as body weight, blood pressure, or calculable biomarkers like HbA1c and LDL cholesterol. It is now equally important to also assess non-traditional risk factors, including systemic inflammation, oxidative stress, and endothelial dysfunction in our patients, regardless of their diabetic status or BMI.

Despite extensive research into GLP-1 receptor agonists—including their mechanisms of action, systemic effects, and potential for weight loss and glycemic control—many questions remain regarding why they exert their effects in the way they do, particularly with respect to cardiovascular benefits and their role in heart failure, especially HFpEF. Future research should be focused on their mechanisms of action since direct studies from human individuals are lacking and all the hypotheses remain promising but inconclusive.

Their metabolic effect and the clinical impact on HF is often closely linked to SGLT2 inhibitors. However, mechanistic differences exist between these two drugs:

On the one hand, SGLT2 inhibitors improve cardiac structure and function by reducing myocardial fibrosis, decreasing ventricular wall stress, and enhancing diuresis and natriuresis. They also exert beneficial renal effects and modulate energy metabolism.

On the other hand, GLP1-RAs have a different profile of action by enhancing glycemic control, promoting weight loss, and modulating autonomic tone and natriuretic peptides, as well as inflammation and endothelial function. Their effects on cardiac remodeling are modest compared to SGLT2 inhibitors, and they are not recommended for reducing heart failure events in current guidelines.

While both drug classes provide cardiometabolic benefits, their primary mechanisms differ and complement distinct aspects of cardiovascular physiology. SGLT2 inhibitors exert pronounced hemodynamic effects through diuresis, natriuresis, and reductions in ventricular wall stress, whereas GLP-1RAs predominantly confer atherogenic and anti-inflammatory benefits, which may indirectly support cardiac health but with less impact on hemodynamics. Together, these complementary profiles highlight the distinct yet potentially synergistic roles of SGLT2 inhibitors and GLP-1RAs in cardiometabolic therapy.

Despite these mechanistic and clinical insights, interpretation of the current data is limited by the design of most studies. The heterogeneity of results across trials may largely reflect the absence of studies specifically designed to evaluate heart failure outcomes. Most of the available evidence comes from trials conducted in populations with type 2 diabetes or obesity, in which heart failure outcomes were typically exploratory or secondary. Importantly, few studies have established a clear definition of heart failure phenotypes, and left ventricular ejection fraction (LVEF) was not consistently or accurately recorded. As a result, it remains unclear which patient subgroups may derive the greatest benefit from GLP-1RA therapy.

Furthermore, the primary endpoints in these trials were often soft outcomes—such as quality of life, exercise capacity, or six-minute walk distance—rather than hard cardiovascular endpoints like HF hospitalization or mortality. The inclusion of heterogeneous populations and trial designs further complicates the interpretation and generalization of these findings. Distinct pathophysiological differences between HFpEF and HFrEF also raise concerns regarding the applicability of results across HF phenotypes.

In addition, given the lack of dedicated trials, important questions remain regarding the long-term cardiovascular benefits of GLP-1RAs, particularly in the context of concurrent use of other cardioprotective agents such as SGLT2 inhibitors or following treatment discontinuation. There may be a potential weight regain and loss of glycemic control upon withdrawal, highlighting the need to assess possible rebound effects in patients with heart failure. Long-term data are needed to determine the durability of benefit, the optimal treatment duration, and strategies for maintaining improvements once therapy is stopped.

Ongoing trials are designed to address these open questions. The CAMEO-SEMA (NCT05371496) study (Cardiac and Metabolic Effects of Semaglutide in Heart Failure) is evaluating semaglutide in patients with obesity-related HF, with a focus on cardiac structure, function, and symptom burden. Finally, SELECT-LIFE (NCT04972721) is an ongoing, long-term observational extension of the SELECT trial, designed to monitor health outcomes for up to 10 years after trial completion, thereby providing valuable data on the durability and safety of GLP-1RA therapy in individuals with cardiometabolic disease. Collectively, these studies will help define whether GLP-1RAs can be incorporated into HF-specific therapeutic strategies and in which patient subgroups their use may be most beneficial.

In parallel, the safety profile of GLP-1RAs must be carefully considered. Commonly reported adverse effects include nausea, vomiting, diarrhea, and abdominal pain. GLP-1RAs have also been associated with an increased risk of gallbladder disease, gastroesophageal reflux, and, rarely, gastroparesis. Rare but serious adverse events include pancreatitis and potential worsening of diabetic retinopathy. Recently, an emerging signal has suggested a potential association with non-arteritic anterior ischemic optic neuropathy (NAION). While causality remains unproven, this finding has raised concerns in certain populations and warrants further investigation [81,82].

## 7. Conclusions

GLP-1 receptor agonists have emerged as promising agents in the management of cardiometabolic diseases.

Current evidence suggests that GLP-1RAs improve quality of life, exercise capacity, and cardiac remodeling and reduce worsening HF events, as well as the composite of CV death or worsening HF in the HFpEF population, emerging as a promising adjunct therapy.

In contrast, their use in patients with HFrEF is not supported due to safety concerns, including higher rates of HF hospitalization and all-cause mortality.

Ongoing and future trials are needed to further define their role across different heart failure phenotypes.

## Figures and Tables

**Figure 1 biomolecules-15-01342-f001:**
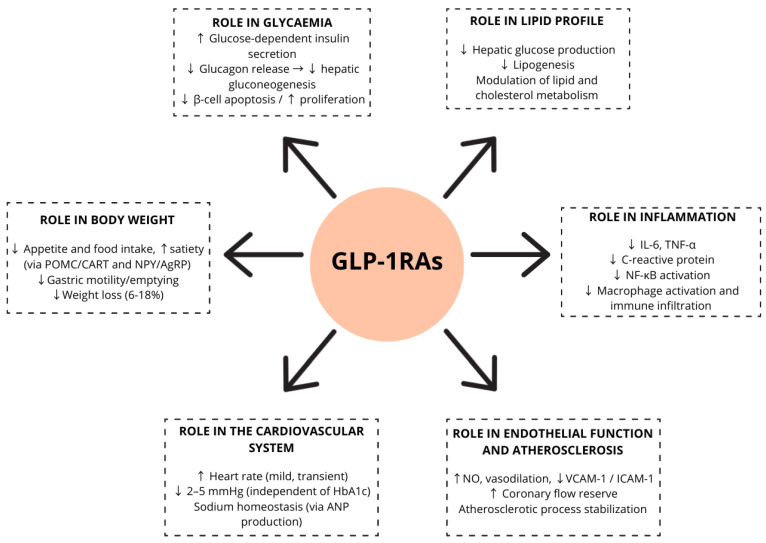
Overview of the mechanisms of action of GLP-1 receptor agonists (GLP-1RAs), including effects on pancreatic hormone secretion, gastric emptying, appetite regulation, body weight, lipid metabolism, inflammation, blood pressure, endothelial function, and cardiovascular system.

**Table 1 biomolecules-15-01342-t001:** GLP-1RAs Cardiovascular Outcome Clinical Trials in type 2 Diabetes Mellitus.

	ELIXA (n = 6068)	LEADER (n = 9340)	SUSTAIN-6 (n = 3297)	EXSCEL (n = 14,752)	Harmony Outcomes (n = 9463)	REWIND (n = 9901)	PIONEER 6 (n = 3183)	AMPLITUDE-O (n = 4076)
Drug	Lixisenatide	Liraglutide	Semaglutide	Exenatide	Albiglutide	Dulaglutide	Semaglutide	Efpeglenatide
Age	60.6 (±9.6)	64 (±7.2)	64.6 (±7.4)	62.0 (±6)	64 (±8.7)	66.2 (±6.5)	66 (±7)	64.5 (±8.2)
Sex (male)	4207 (69%)	6003 (64%)	2002 (61%)	9149 (62%)	6569 (69%)	5312 (54%)	2176 (68%)	2732 (67%)
Follow-up (median, years)	2.1	3.8	2.1	3.2	1.6	5.4	1.3	1.81
BMI (kg/m^2^)	30.1 (±5.6)	32.5 (±6.3)	32.8 (±6.2)	32.7 (±6.4)	32.3 (±5.9)	32.3 (±5.7)	32.3 (±6.5)	32.7 (±6.2)
HbA1c, %	7.7 (±1.3)	8.7 (±1.6)	8.7 (±1.5)	8.1 (±1.0)	8.7 (±1.5)	7.3 (±1.1)	8.2 (±1.6)	8.9 (±1.5)
eGFR, mL/min per 1.73 m^2^	78 (±21)	80 (NR)	80 (61–92)	77 (61–92)	79 (±25)	77 (±23)	74 (±21)	72 (±22)
Previous CVD, %	100%	81%	83%	73%	100%	31%	85%	90%
HF history, %	22%	18%	24%	16%	20%	9%	NR	18%
SGLT2i use, %	NR	NR	<1%	1%	6%	<1%	10%	15%
3-point MACE	1.02 (0.89–1.17)	0.87 (0.78–0.97)	0.74 (0.58–0.95)	0.91 (0.83–1.00)	0.78 (0.68–0.90)	0.88 (0.79–0.99)	0.79 (0.57–1.11)	0.73 (0.58–0.92)
CV death	0.98 (0.78–1.22)	0.78 (0.66–0.93)	0.98 (0.65–1.48)	0.88 (0.76–1.02)	0.93 (0.73–1.19)	0.91 (0.78–1.06)	0.49 (0.27–0.92)	0.72 (0.50–1.03)
Nonfatal MI	1.03 (0.87–1.22)	0.86 (0.73–1.00)	0.74 (0.51–1.08)	0.97 (0.85–1.10)	0.75 (0.61–0.90)	0.96 (0.79–1.16)	1.18 (0.73–1.90)	0.78 (0.55–1.10)
Nonfatal Stroke	1.12 (0.79–1.58)	0.86 (0.71–1.06)	0.61 (0.38–0.99)	0.85 (0.70–1.03)	0.86 (0.66–1.14)	0.76 (0.61–0.95)	0.74 (0.35–1.57)	0.80 (0.48–1.31)
HF hospitalizations	0.96 (0.75–1.23)	0.87 (0.73–1.05)	1.11 (0.77–1.61)	0.94 (0.78–1.13)	NA	0.93 (0.77–1.12)	0.86 (0.48–1.55)	0.61 (0.38–0.98)
All-cause mortality	0.94 (0.78–1.13)	0.85 (0.74–0.97)	1.05 (0.74–1.50)	0.86 (0.77–0.97)	0.95 (0.79–1.16)	0.90 (0.80–1.01)	0.51 (0.31–0.84)	0.78 (0.58–1.06)

Data are mean (SD), n (%), or median (IQR), unless otherwise specified. eGFR: estimated glomerular filtration rate. NR/NA: not reported/assessed. This table has been adapted from Wilcox et al. (2020) [9] and Kristensen et al. (2019) [53] and expanded to include additional cardiovascular and renal outcome data from the AMPLITUDE-O trial (Gerstein et al., 2021) [52].

**Table 2 biomolecules-15-01342-t002:** Clinical Trials in patients with heart failure.

	STEP HFpEF (n = 529)	STEP HFpEF DM (n = 616)	SUMMIT (n = 731)	FIGHT (n = 300)	Lepore et al. (n = 82)	LIVE (n = 241)
Drug	Semaglutide	Semaglutide	Tirzepatide	Liraglutide	Albiglutide	Liraglutide
Age	69 (62–75)	69 (62–74)	65.5 (±10.5)	61 (52–68)	56 (±10)	65 (±9.2)
Sex (male)	43.9%	58.7%	45.1%	80%	74%	89%
Follow-up (weeks)	52	52	104	25	12	24
BMI (kg/m^2^)	37.0 (33.7–41.4)	36.9 (33.6–41.5)	38.3 (±6.4)	32 (26–37)	31 (±7)	28 (±3.8)
T2D (%)	0%	100%	47.8%	59%	0%	32%
eGFR, mL/min/m^2^	NR	NR	64.5 (±23.7)	NR	NR	79 (±20)
NT-proBNP, pg/mL	450.8 (218.2–1015.0)	477.8 (251.2–969.2)	196 (56–488)	1936 (1075–4231)	89.6 (±23.2) ^•^	413 (208–926)
LVEF, %	57.0 (50.0–60.0)	57.0 (50.0–61.0)	61.0 (6.5)	25% (20–33)	31% (1.6)	33.7% (7.6)
Change in Quality of Life *	7.8 (4.8 to 10.9)	7.3 (4.1 to 10.4)	6.9 (3.3 to 10.6)	0.6 (−4.5 to 5.8)	2.5 (4.8), *p* = 0.61	−1.6 (−5.3, 2.0)
Body weight loss, % ^††^	–10.7 (–11.9 to –9.4)	–6.4 (–7.6 to −5.2)	−11.6 (−12.9 to −10.4)	−1.8 (−3.9 to 0.3)	−1.6 (0.4), *p* = 0.003	−0.8 (−1.1, −0.4)
Change in 6-MWT distance	20.3 (8.6 to 32.1)	14.3 (3.7 to 24.9)	18.3 (9.9 to 26.7)	5 (−29 to 39)	9 (±16), *p* = 0.58	24 (2 to 47), *p* = 0.04
HF event	0.08 (0.00 to 0.42) **	0.40 (0.15 to 0.92) **	0.62 (0.41 to 0.95) ^†^	146 vs. 156, *p* = 0.31 ^¶^	NA	NA

Data are mean (SD) or median (IQR), unless otherwise specified. eGFR: estimated glomerular filtration rate. NR/NA: not reported/assessed. This table summarizes data extracted from the original trials cited in the text, including STEP-HFpEF, STEP-HFpEF DM, SUMMIT, FIGHT, Lepore et al., and LIVE. BMI: Body Mass Index; T2D: Type 2 Diabetes; eGFR: estimated glomerular filtration rate; LVEF: Left Ventricle Ejection Fraction; KCCQ-CSS: Kansas City Cardiomyopathy Questionnaire Clinical Summary Score; 6-MWT: 6-min walking test. * STEP HFpEF, STEP HFpEF DM, SUMMIT and FIGHT trial assessed quality of life with KCCQ-CSS. Lepore et al. and LIVE trial used Minnesota Living with Heart Failure Questionnaire. ** HF event was a composite of hospitalization or urgent visit for heart failure (which required intravenous therapy). ^†^ HF event was a composite of hospitalization for HF, intravenous drugs in an urgent care setting, or intensification of oral diuretic therapy. ^††^ Change in body weight was assessed in STEP HFpEF, STEP HFpEF-DM and SUMMIT as the percentual change of weight loss, whereas in FIGHT, Lepore and LIVE was net change in BMI. ^¶^ HF event was a global rank score across 3 hierarchical tiers: time to death, time to rehospitalization for heart failure, and time-averaged proportional change in N-terminal pro-B-type natriuretic peptide level from baseline to 180 days. ^•^ In Lepore et al, the results are expressed for Brain Natriuretic Peptide (in ng/L).

## Data Availability

Data sharing is not applicable to this article as no new data were created or analyzed in this study.

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
