# Peer review of "Role of Glucagon-like Peptide-1 Receptor Agonists (GLP-1RAs) in Patients with Chronic Heart Failure"

_biomolecules, 2025, doi:10.3390/biom15091342_

Round 1
Reviewer 1 Report
Comments and Suggestions for Authors
The present review represents an innovative contribution to the scientific literature for several reasons. While most previous studies and reviews on GLP-1 receptor agonists (GLP-1RAs) have focused on traditional cardiovascular endpoints such as MACE, this review specifically addresses the role of GLP-1RAs in chronic heart failure (HF). In particular, the manuscript provides a detailed analysis of GLP-1RA effects in patients with both HFpEF and HFrEF; integrates data from clinical trials, preclinical studies, and molecular mechanisms, offering a comprehensive and multidimensional perspective and specifically considers heart failure hospitalization as a distinct endpoint, which is often underrepresented in major CVOTs.
Major comments
-The Authors might consider including a summary image of the mechanisms of action of GLP-1RAs in Section 2.
-I would suggest that the Authors add a paragraph summarizing the pharmacological treatments currently recommended for HF. It would be particularly useful to highlight whether current guidelines already distinguish between therapeutic approaches for HFrEF and HFpEF, as this would provide important clinical context for the discussion.
-The section on HFpEF appears more detailed and better supported by recent. In contrast, the HFrEF part is less comprehensive. The Authors should balance the discussion by further elaborating on available (even negative or inconclusive) data in HFrEF.
-It would be useful to briefly describe the methodology of the literature search (databases, timeframe, keywords, inclusion criteria).
Minor comments
-The Authors could better clarify whether the increased risk of heart failure–related hospitalization and mortality is more strongly associated with type 2 diabetes or with obesity.
-The Authors could clarify whether there are other antidiabetic drugs, besides SGLT2 inhibitors, that have shown beneficial effects in reducing heart failure–related hospitalizations or mortality.
-Lines 66-68: This review clearly and thoroughly illustrates the gap in current knowledge regarding the role of GLP-1RAs in HF, but it does not aim to fill it. Rather, it sharply highlights an important problem in the clinical management of this patient population. I would suggest that the Authors reconsider this sentence.
-The Authors should add a short section summarizing ongoing clinical trials with GLP-1RAs in HF.
-Minor stylistic/grammar corrections could improve readability (e.g., verb tenses when discussing past vs ongoing studies).
Reviewer 2 Report
Comments and Suggestions for Authors
Current review article discussed the role of Glucagon-like peptide-1 receptor agonists (GLP-1RAs) in heart failure (HF). Please conduct the concerns below.
- HF with preserved ejection fraction (HFpEF) has been the target in current report. It needs rationale background(s).
- Source of the data in Figure 1 must show in the figure legend. Same for Figure 2 in addition.
- In basic studies, AMPK has been suggested as the signal of GLP-1RAs for HF. GLP-1RAs may activate AMPK to link autophagy that is contraindicated in HF. This view seems helpful to conduct with the data in current report.
- Novelty was not described in conclusion.
- Compare to cardioprotective agents such as SGLT2 inhibitors, limitation(s) of GLP-1RAs will be helpful.
Round 2
Reviewer 2 Report
Comments and Suggestions for Authors
It has been improved according to comments. Thank you very much.